# Improving Zero-Shot Generalization of Instruction Tuning by Data Arrangement

## Abstract

Understanding alignment techniques begins with comprehending zero-shot generalization brought by instruction tuning, but little of the mechanism has been understood. Existing work has largely been confined to the task level, without considering that tasks are artificially defined and, to LLMs, merely consist of tokens and representations. To bridge this gap, we investigate zero-shot generalization from the perspective of the data itself. We first demonstrate that zero-shot generalization happens very early during instruction tuning, with loss serving as a stable indicator. Next, we investigate the facilitation of zero-shot generalization by data arrangement through similarity and granularity perspectives, confirming that encountering highly similar and fine-grained training data earlier during instruction tuning, without the constraints of defined "tasks", enables better generalization. Finally, we propose a more grounded training data arrangement method, Test-centric Multi-turn Arrangement, and show its effectiveness in promoting continual learning and further loss reduction. For the first time, we show that zero-shot generalization during instruction tuning is a form of similarity-based generalization between training and test data at the instance level. We hope our analysis will advance the understanding of zero-shot generalization during instruction tuning and contribute to the development of more aligned LLMs.

## 1 Introduction

The extraordinariness of large language models (LLMs) was originally brought by the zero-shot generalization of instruction tuning (Brown et al., 2020). Early studies have found that when diverse prompts are added to the inputs of traditional natural language processing (NLP) tasks and fed into the model for instruction tuning, the model can generalize to tasks it has never encountered before (Chung et al., 2024; Longpre et al., 2023; Sanh et al., 2021; Wang et al., 2022; Wei et al., 2021). To date, instruction tuning (Chung et al., 2024; Sanh et al., 2021; Wei et al., 2021) has become a crucial phase in LLM training, often preceding methods that incorporate preference data. In the meantime, the concept of "task" is also becoming increasingly blurred. Researchers are no longer constructing instruction data in the ways traditional NLP tasks dictate, but rather, they hope these tasks will be as close to reality and as diverse as possible (Chiang et al., 2023; Ding et al., 2023; Rajani et al., 2023; Taori et al., 2023; Zhao et al., 2024b).

Although nearly all LLMs benefit from the zero-shot generalization brought about by instruction tuning, the in-depth and fine-grained research on this phenomenon is still insufficient. Particularly, there are few accurate and comprehensive conclusions about when and in what form it occurs, and how it would be influenced at an instance level. A line of existing research focuses on understanding the relationships in task-pair transfer (Song et al., 2019; Vu et al., 2020; Zamir et al., 2018), suggesting that not all tasks contribute positively to zero-shot generalization, with some may even result in negative transfer effects (Kim et al., 2023; Muennighoff et al., 2023; Zhou et al., 2022). These works tend to train on one task and then evaluate on another (Kim et al., 2023; Zhou et al., 2022), or simply calculate intermediate task (Vu et al., 2020) or instruction (Lee et al., 2024) transfer scores. However, all these efforts to select the most informative tasks to improve zero-shot generalization are *restricted to the task-based framework*. This approach assumes that human-defined "tasks" and even "categories" are sufficiently reasonable, but this is often not the case, as gaps exist regarding how humans and LLMs perceive the instruction tuning data. To this end, we strive to break free from the task-level framework and explore "generalization" from a more granular temporal perspective.

In this paper, we conduct a comprehensive investigation of the zero-shot generalization during instruction tuning and attempt to answer some critical questions: In instruction tuning, i) *when does zero-shot generalization occur?* ii) *how can we more accurately understand the role of data in zero-shot generalization?* iii) *how can we effectively improve zero-shot generalization?*

To answer the first research question, we attempt to pinpoint the timing of zero-shot generalization during instruction tuning. We discover that zero-shot generalization occurs extremely early, with only about 160 random data points needed for significant generalization. This indicates that LLM's instruction-following ability can be effectively unlocked by leveraging merely a few sample training data, similar to existing works (Chen et al., 2023; Gudibande et al., 2023; Zhao et al., 2024a; Zhou et al., 2024) but we provide a more granular analysis. In addition, we take a step forward by demonstrating that loss serves as a stable and fair indicator of zero-shot generalization during our measurement, offering a more consistent measure than traditional metrics such as the ROUGE series, Exact-Match, and Reward Model scores. (Section 2)

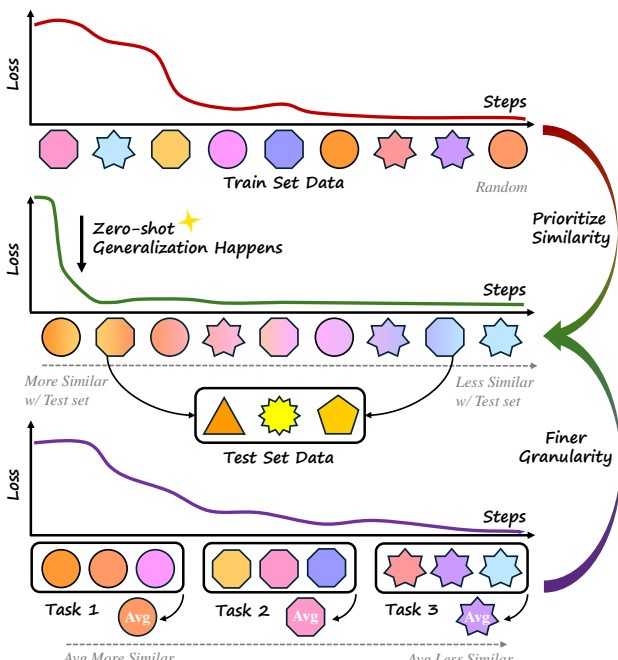

Figure 1: An overview of data arrangement framework: each graphic represents a data, and each shape represents a distinct task. The more similar the color, the closer the distance between data. Prioritizing similar, fine-grained data early in instruction tuning, rather than adhering strictly to task boundaries, leads to better zero-shot generalization, as indicated by the rapid decrease in loss.

To further investigate why zero-shot generalization occurs early, we examine the generalization performance of each individual test task (a total of 225 tasks) by analyzing the changes in loss, rather than assessing overall generalization. We find that under different training data arrangements, the same test task can exhibit varying generalization behaviors. It may generalize rapidly in the early stages, while at other points, it can show slower or delayed generalization, or even fail to generalize altogether, despite an overall tendency for early generalization. To understand the role of data in this process, we identify two perspectives: **similarity** and **granularity**.

*From a similarity perspective*, we discover that the model's generalization is not truly "zero-shot", as a high resemblance between the training and test data distributions could significantly impact generalization. To quantify similarity in experiments, we perform a linear combination of the average and minimum cosine similarity between the training and test data, revealing that encountering training data with high similarity early during instruction tuning improves zero-shot generalization. *From a granularity perspective*, we disclose that artificially defined "tasks" are not suitable for measuring generalization. Instead, the similarity measure at the instance level serves as a better and more essential indicator. We reveal through experiments that, by treating all data points equally without the constraints of pre-defined tasks, we can better improve zero-shot generalization. As shown in Figure 1, encountering highly similar and fine-grained training data earlier during instruction tuning, without the constraints of defined "tasks", enables better generalization. (Section 3)

Though the weighted similarity measure we adopt is effective in improving zero-shot generalization, it still suffers from inherent limitations in that the individual role of each test data point cannot be distinguished, as the whole test set is considered holistically during the calculation of similarity. To this end, we propose the Test-centric Multi-turn Arrangement (TMA). TMA arranges training data according to each individual test data instance and seeks to improve zero-shot generalization regardless of the presence of predefined tasks. We demonstrate through experiments the effectiveness

of TMA and unveil that early access to data of higher similarity during instruction tuning can facilitate continual learning and further loss reduction. (Section 4)

We summarize our contributions as follows:

- We show that zero-shot generalization occurs at the very early stage during instruction tuning, while loss serves as a more stable and fair metric compared to traditional ones.
- We identify *similarity* and *granularity* as two perspectives to gain a deeper understanding of zero-shot generalization, revealing that encountering highly similar and fine-grained training data earlier during instruction tuning enables better generalization.
- We propose the Test-centric Multi-turn Arrangement, a more grounded training data arrangement method, and show that accessing high-similarity data during instruction tuning can facilitate continual learning and further loss reduction.

## 2 POSITIONING ZERO-SHOT GENERALIZATION

Early research shows that instruction tuning, which applies to various NLP tasks formatted with instructions, can generalize to various unseen tasks. However, most studies (Chung et al., 2024; Iyer et al., 2022; Longpre et al., 2023) focus on integrating diverse tasks or instruction templates, using human-generated or synthetic data, or exploring different fine-tuning strategies, while few studies address the timing of zero-shot generalization. To bridge this gap, we first seek to identify **when zero-shot generalization actually occurs during instruction tuning**, and then justify that loss serves as a more stable and fair metric to measure zero-shot generalization compared to traditional ones including ROUGE series, Exact-Match, Reward Model scores, etc. We begin by giving a formalization of zero-shot generalization.

**Formalization.** In multi-task scenarios, zero-shot generalization refers to the ability to perform effectively on unseen tasks ($\mathcal{T}_{\text{Unseen}}$), while only trained on a subset of tasks ($\mathcal{T}_{\text{Seen}}$). For each task $T \in \mathcal{T}_{\text{Seen}} \cup \mathcal{T}_{\text{Unseen}}$, there exists an instructional description $I_T$, as well as several instances, where each instance is composed of an input $x_{T,i}$ and an output $y_{T,i}$. We define a model $M$ as capable of generalization on unseen tasks if, after training on every task in $\mathcal{T}_{\text{Seen}}$, given an unseen task $T \in \mathcal{T}_{\text{Unseen}}$, and for any $(x_{T,i}, y_{T,i})$, the model's output $\hat{y} = M(I_T, x_{T,i})$ and the label $y_{T,i}$ achieve a score surpassing a certain threshold, with regard to the selected metrics, indicating successful generalization on the task $T$.

### 2.1 OCCURRENCE OF ZERO-SHOT GENERALIZATION IN INSTRUCTION TUNING

From the formalization above, it is clear that the measurement of zero-shot generalization is largely dependent upon the selection of metrics. However, the impact of various metrics on zero-shot generalization is rarely studied. To this end, we first evaluate multiple metrics to see if they are suitable for zero-shot generalization and demonstrate that:

> 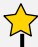 **Takeaway 1**: Zero-shot generalization occurs during the very early stage of instruction tuning, despite the metrics chosen for measurement.

**Data and Settings.** We utilize three multi-task datasets, namely Natural Instructions V2 (NIV2) (Wang et al., 2022), Public Pool of Prompts (P3) (Sanh et al., 2021), and Flan-mini (Ghosal et al., 2023), for our analysis. For NIV2, we utilize the default track and training-test split for instruction tuning and evaluation. For P3, we employ training and test tasks consistent with the vanilla T0 model[1]. For Flan-mini, we randomly partition the training and test tasks. We choose pre-trained LLaMA-2-7B (Touvron et al., 2023) as our base model. For other details including dataset, hyper-parameters, and prompt template, please refer to Appendix A. All subsequent experiments are based on this setup, where we adopt to save a series of full-parameter fine-tuning checkpoints and evaluate each on the test set to observe the results regarding specified metrics.

**Metrics.** We experiment with multiple metrics, including Exact-Match, ROUGE-1, ROUGE-L, and RM score, to test their ability to reasonably reflect zero-shot generalization. For P3 and Flan-mini ,

---

[1] https://huggingface.co/bigscience/T0pp

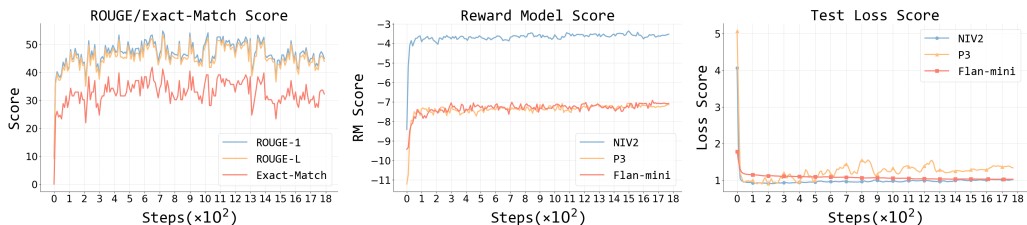

Figure 2: Average ROUGE-1, ROUGE-L, and Exact-Match scores (left), average RM scores (middle), and average loss scores (right) of checkpoints fine-tuned on NIV2 (left, middle, right), P3 (middle, right), and Flan-mini (middle, right), all evaluated on unseen tasks.

Exact-Match is commonly applied in previous studies due to its simplicity, while NIV2 additionally incorporates ROUGE-1 and ROUGE-L as metrics. Besides, in reinforcement learning scenarios, the reward model (RM) often plays a vital role (Cui et al., 2023; Yuan et al., 2024) and serves as a proxy for human preferences. This makes the RM score also a plausible metric to reflect zero-shot generalization. Empirically, we use UltraRM-13B (Cui et al., 2023) as the reward model when measuring RM score.

**Results.** We demonstrate that zero-shot generalization occurs at a very early stage during instruction tuning regardless of metric choice. As depicted in the left plot of Figure 2, using ROUGE-1, ROUGE-L, and Exact-Match as metrics, the scores rise from approximately 15 to over 35 in just about 10 training steps, indicating significant generalization with only around 160 training samples in our setting. In the middle plot, the RM score exhibits a similar trend, stabilizing around 50 steps across all three datasets.

Despite the similar trend they present, it should be noted that ROUGE-1, ROUGE-L, and Exact-Match as metrics all entail the resulting curves being seriously unstable, while the RM score for NIV2 is significantly higher than those for the other two datasets, indicating a certain bias induced (more details discussed in Appendix A.5). This leads us to seek a more reasonable metric as an indicator to evaluate zero-shot generalization.

## 2.2 Loss as the Measurement for Zero-Shot Generalization

Loss is commonly applied across model pre-training and fine-tuning scenarios. For example, the scaling law (Clark et al., 2022; Henighan et al., 2020; Kaplan et al., 2020) entails predicting loss based on model parameter count and dataset size, unsupervised learning uses cross-entropy loss to quantify the difference between probability distribution, etc. Recent studies have also delved into understanding emergent abilities from the perspective of loss, indicating that when pre-training loss drops below a specific threshold, the model can perform well on downstream tasks (Du et al., 2024). All these measures suggest loss to be a promising metric for evaluating zero-shot generalization. Therefore, we comprehensively study and justify that:

> 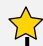 **Takeaway 2**: Loss serves as a stable and reasonable metric to measure zero-shot generalization due to its stability and fairness across datasets.

**Data and Settings.** We use the same dataset as in the previous experiment and generate outputs for sampled test data points using a series of instruction-tuned checkpoints we derived. We then calculate the average cross-entropy loss against the corresponding labels within each step. Please refer to Appendix A.4 for more details.

**Results.** Zero-shot generalization similarly occurs at an early stage of instruction tuning with loss as the metric. As shown in the right plot of Figure 2, all three datasets reach their lowest points in terms of loss within less than 50 steps, which strengthens the conclusion that zero-shot generalization occurs early.

Moreover, compared to the left and middle plots in Figure 2, it is noteworthy that loss as an indicator is more stable and fair across different datasets, entailing it as a more reasonable metric for the

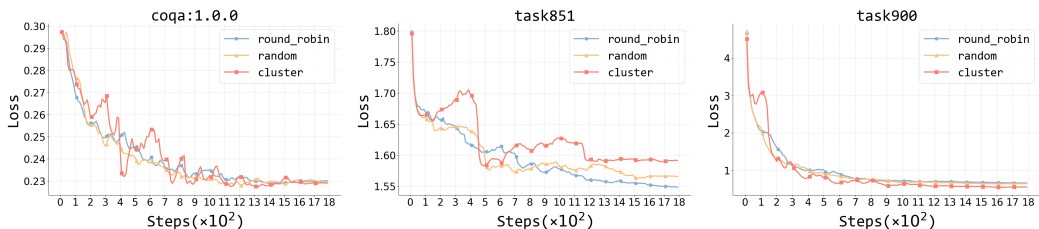

Figure 3: Sudden decrease in the average loss under cluster scheduling for the three tasks at steps 400, 450, and 150 respectively.

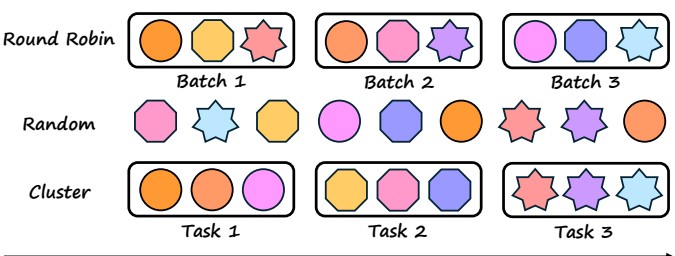

Figure 4: An overview of Round Robin, Random and Cluster data arrangements. Definitions of colors and shapes are consistent with those in Figure 1.

measurement. We also provide a case study of loss curves with regard to different unseen tasks, as detailed in Appendix A.6.

## 3 FACILITATING ZERO-SHOT GENERALIZATION

Acknowledging the importance of metrics in measuring the positioning of zero-shot generalization, we next seek to investigate **why generalization occurs at an early stage and what role training data plays during this phase**. Our initial focus lies in the analysis of how various simple training data arrangements affect zero-shot generalization in a fine-grained manner. Then, we investigate the facilitation of zero-shot generalization from both data *similarity* and *granularity* perspectives.

### 3.1 EFFECT OF TRAINING DATA ARRANGEMENTS

The model receives only a limited amount of data at the early stage of instruction tuning. Therefore, despite the scarcity, these data ought to play a significant role in facilitating generalization. Guided by this intuition, we explore the impact of exposure to different training data arrangements from a temporal perspective.

**Data and Settings.** We apply 1600 Flan-mini training tasks to get a series of instruction-tuned checkpoints and evaluate them on various unseen test tasks in Flan-mini. As shown in Figure 4, we examine the following three training data arrangements:

- **Round Robin**: We select one data point from each task to form a data batch, ensuring that training tasks are as evenly distributed as possible.
- **Cluster**: We arrange all data from each task together, resulting in task-level clusters throughout the entire training dataset.
- **Random**: We randomly shuffle all training data as a baseline for comparison.

**Results.** Different training data arrangements entail different loss curve patterns. As shown in Figure 3, the pattern of random and round-robin scheduling is similar due to round-robin being an extreme form of data shuffling. However, cluster scheduling differs largely from both. At certain steps during the instruction tuning, there exists a sudden decrease in average loss across different

test tasks. This further highlights that leveraging a relatively small amount of data may induce a substantial drop in the loss, and the same test task can exhibit varying generalization behaviors under different training data arrangements.

### 3.2 Zero-Shot Generalization through Data Similarity and Granularity

We have observed that training data arrangements could lead to significant changes in the loss curve and that the timing of presence for certain data may greatly facilitate generalization on unseen tasks. With these findings, we naturally ask what is the best arrangement that facilitates zero-shot generalization the most, and how to arrange these "certain data" that improve early generalization. In the following subsections, we seek to address these questions through two perspectives: **similarity** and **granularity**.

#### 3.2.1 Improving Zero-Shot Generalization through High-Similarity Data

Previous research (Dai et al., 2019; Yauney et al., 2023) has consistently demonstrated that the performance of downstream tasks improves when the similarity between the pre-training data and the downstream task data increases. This finding aligns with our intuitive understanding that data points with higher similarity can better facilitate generalization. Based on these insights, we propose and subsequently validate that:

> 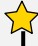 **Takeaway 3**: Encountering data with high similarity early during instruction tuning will greatly improve zero-shot generalization.

**Selection of Similarity Measures.** To validate our hypothesis, we first define what similarity measures to apply. Based on our setting and previous works, we investigate two main categories of similarity measures:

- **N-gram similarity**: We respectively measure the similarity distance by calculating the KL divergence between the bigram word distributions of the training set data and test set data.
- **Embedding similarity**: We utilize all-MiniLM-L6-v2 [1] from Sentence Transformer (Reimers & Gurevych, 2019) to compute embeddings for each training and test data and then calculate the Cosine and Euclidean similarity distances, as detailed in Appendix B.3. Next, we refer to four classical distance calculation methods [2], namely "max" (maximal distance), "min" (minimal distance), "avg" (unweighted average distance), and "centroid" (centroid distance), to represent the distance from the training set data to test set data.

We analyze a series of instruction-tuned checkpoints on Flan-mini and calculate the similarity measure score between the training data seen by the $k^{th}$ checkpoint and all the test data. This entails in total nine similarity calculation methods ({N-gram} + {Cosine, Euclidean} × {max, min, avg, centroid}). Please refer to Appendix B.3 for calculation details. Through analyzing Figure 15, we find that i) Cosine and Euclidean measures show little difference; ii) the drop in minimum distance calculation coincides with the drop in loss curve; iii) average distance calculation promotes fairness by involving all test data in the actual calculation. Therefore, we consider using a simple linear combination of Cosine Average (Cosine-Avg) and Cosine Minimum (Cosine-Min) for similarity calculation in later experiments to raise the stability of computation and leverage the strengths from both Cosine-Avg and Cosine-Min. We also prove they both satisfy optimal substructure, ensuring that the effect of training set arrangement can accumulate over time as more data point is presented to the model with good scalability, detailed in Appendix B.5.

**Data and Settings.** We utilize the Flan-mini dataset and randomly sample up to 20 instances for each training task. Each test task consists of at most five test data points to form a test set. For each training data point $x_i$, we calculate the similarity measure score between $x_i$ and the whole test set $\mathcal{D}_{\text{Test}}$. We arrange the training data based on this score. Specifically, we examine three training arrangements: Nearest First Training (NFT), Farthest First Training (FFT), and Random Training (RT), as shown in Figure 5. This setup allows us to differentiate between the nearest and farthest data points in terms of

---

[1] https://huggingface.co/sentence-transformers/all-MiniLM-L6-v2
[2] https://en.wikipedia.org/wiki/Hierarchical_clustering

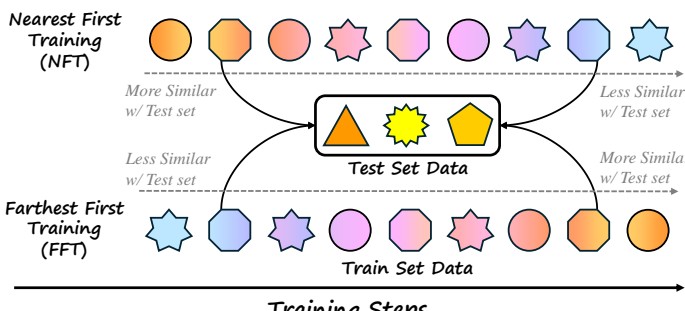

Figure 5: An overview of NFT and FFT data arrangements. Definitions of colors and shapes are consistent with those in Figure 1.

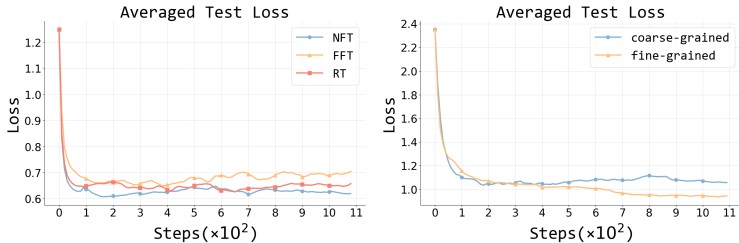

Figure 6: **Left**: The impact of the three similarity settings (NFT, FFT, and RT) on averaged test loss. **Right**: The impact of different granularity settings on averaged test loss.

the temporal dimension of instruction tuning. We perform instruction tuning on the three training data arrangements, resulting in a series of fine-grained checkpoints. And then calculate the average loss for each checkpoint on the test set containing various test tasks.

**Results.** The earlier the model encounters data with high similarity to the test set, the more beneficial it is for zero-shot generalization. As shown in the left plot of Figure 6, we can observe that the NFT setting exhibits a rapid and low loss reduction, indicating better zero-shot generalization. In contrast, the FFT setting shows relatively poorer zero-shot generalization compared to the baseline RT setting.

### 3.2.2 IMPROVING ZERO-SHOT GENERALIZATION THROUGH FINE-GRAINED DATA

Traditional methods to improve zero-shot generalization are mostly confined to the task level, focusing on task-pair transfer. However, the so-called "tasks" or "categories" are artificially defined and, from the perspective of LLMs, they are merely a collection of tokens or embedding representations. Therefore, different "tasks" or "categories" may still appear relatively similar to LLMs, while instances from the same task may exhibit profound differences. Thus, we propose and validate that:

> 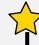 **Takeaway 4**: Treating all data points equally in finer granularity without the concept of "task" as constraints better improves zero-shot generalization.

**Data and Settings.** We use the Flan-mini dataset and randomly sample up to 20 instances for each training task. We employ two approaches to arrange the training data: i) **coarse-grained setting**, where all instances under each training task are clustered. We define the embedding of a task as the average embedding of all instances under that task and arrange the clusters based on NFT, as shown in Figure 1. ii) **fine-grained setting**, where all data instances are directly arranged basen on NFT instead of being clustered first.

**Results.** Compared to the coarse-grained setting, the fine-grained setting is more beneficial for improving zero-shot generalization. As shown in the right plot of Figure 6, the loss curve for the fine-grained setting decreases more quickly and effectively, indicating that removing task framework constraints can further improve zero-shot generalization.

## 4 UNDERSTANDING ZERO-SHOT GENERALIZATION

In previous experiments, we empirically apply the linear combination of Cosine-Avg and Cosine-Min. However, these measures inherently possess shortcomings: i) *Cosine-Avg cannot distinguish the variance within the test set*. When the test set collapses to the point where all data points are the same, the distribution becomes a spike; when the test set data is sufficiently spread out, the distribution becomes uniform. In both cases, the Cosine-Avg distance from all test data points to a certain training data point remains consistent, indicating that it could not distinguish the role of each test data point. ii) *Cosine-Min cannot holistically represent all test set data points*. If a certain test data point is closer to all training data points than all other test data points, Cosine-Min only considers the distance to that test data point and does not take into account others, even if they are relatively far away.

Therefore, we seek an approach that can better distinguish and arrange the training data for enhanced zero-shot generalization. To this end, we present the Test-centric Multi-turn Arrangement (TMA).

---

**Algorithm 1** Test-centric Multi-turn Arrangement Method

---

**Require:** Dataset $\mathcal{D}$ split into training set $\mathcal{D}_{\text{train}}$ and test set $\mathcal{D}_{\text{test}}$
**Ensure:** Sub-training sets $\mathcal{D}_{\text{train}}^i$ for each round $i$
1: $i \leftarrow 0$
2: **while** $\mathcal{D}_{\text{train}} \neq \emptyset$ **do**
3:      $i \leftarrow i + 1$
4:      $\mathcal{D}_{\text{train}}^i \leftarrow \emptyset$
5:      **for all** $x \in \mathcal{D}_{\text{test}}$ **do**
6:          Find the **nearest** data point $y \in \mathcal{D}_{\text{train}}$ to $x$ based on cosine similarity
7:          $\mathcal{D}_{\text{train}}^i \leftarrow \mathcal{D}_{\text{train}}^i \cup \{y\}$
8:      **end for**
9:      $\mathcal{D}_{\text{train}} \leftarrow \mathcal{D}_{\text{train}} \setminus \mathcal{D}_{\text{train}}^i$
10: **end while**
11: **return** $\mathcal{Q}_{\text{train}} = \{\mathcal{D}_{\text{train}}^1, \mathcal{D}_{\text{train}}^2, \ldots, \mathcal{D}_{\text{train}}^k\}$

---

**Formalization.** We formalize TMA algorithm in Algorithm 1. This arrangement of the training data ensures that the embedding of each test data point is equally considered, thus taking into account all their characteristics. It avoids the limitations of Cosine-Avg, which disregards the distribution within the test data, or Cosine-Min, which only considers individual test data embeddings. A more detailed investigation of our arrangement method is provided in Appendix C.4.

### 4.1 TEST-CENTRIC MULTI-TURN ARRANGEMENT IMPROVES ZERO-SHOT GENERALIZATION

With all the nice properties presented by TMA, we show that:

> ⭐ **Takeaway 5**: Test-centric Multi-turn Arrangement can further improve zero-shot generalization, regardless of whether the data source involves the concept of "task" or not.

**Data and Settings.** We employ two types of datasets: i) datasets with task splits, such as Flan-mini (Ghosal et al., 2023), and ii) datasets without task splits, such as ShareGPT (Wang et al., 2023) and NoRobots (Rajani et al., 2023). Flan-mini consists of task-specific splits, while ShareGPT and NoRobots are general dialogue datasets. We arrange the training data by applying Algorithm 1 and examine the same three training arrangements, namely NFT, FFT, and RT, which are consistent with the experimental setup in Section 3.2.1. Specifically, NFT under this setting refers to the sequential training data order as returned by Algorithm 1, and FFT refers to its reverse. For detailed configurations, please refer to Appendix C.2.

**Results.** Using our proposed TMA to arrange training data from nearest to farthest significantly improves zero-shot generalization. As illustrated in Figure 7, whether with the task split in Flan-mini (left) or without the task split in ShareGPT (middle) and NoRobots (right), the loss curve under the NFT setting decreases more rapidly while reaching a lower point, whereas the FFT setting results in the poorest performance. This validates the effectiveness of our arrangement method TMA. We also conduct an ablation study on the final distribution of $\mathcal{Q}_{\text{train}}$ returned by Algorithm 1, as detailed

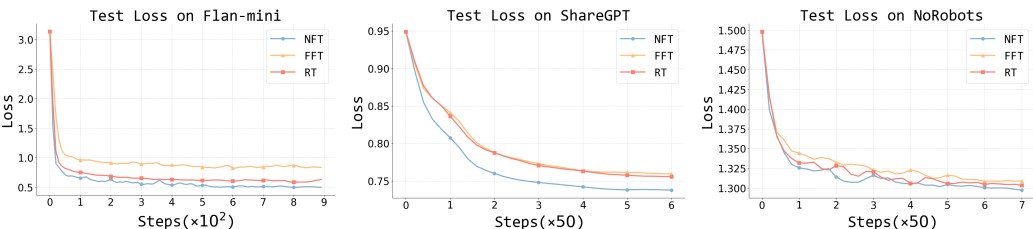

Figure 7: Averaged test loss of three similarity settings (NFT, FFT, and RT) under Test-centric Multi-turn Arrangement on Flan-mini (left), ShareGPT (middle), and NoRobots (right).

in Appendix C.3, and reveal that accessing high-similarity data during instruction tuning can facilitate continual learning and further loss reduction.

### 4.2 EARLY-SELECTED SUB-TRAINING SET SHOWS HIGHER EFFECTIVENESS

The final training set $\mathcal{Q}_{\text{train}}$ in the TMA algorithm is formed through multiple rounds of selections in order. We show that:

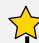 **Takeaway 6**: The earlier the sub-training set is selected by the TMA, the better its effect in improving zero-shot generalization.

**Data and Settings.** We apply Flan-mini as in the previous experiment. Specifically, we cluster the training data selected from rounds $[i, i + 5)(i = 0, 5, 10, 15, 20, 25)$ based on TMA, and respectively perform instruction tuning. Then, for each instruction-tuned checkpoint, we calculate the average loss on the test set.

**Results.** The training data selected from the earliest rounds generally exhibits higher quality. As shown in Figure 8, we observe that from rounds $[0, 5)$ to $[25, 30)$, the decrease in loss becomes slower, and the minimum loss value tends to be higher. This indicates that our arrangement method could successfully distinguish high-quality training data and apply them early in instruction tuning to improve zero-shot generalization.

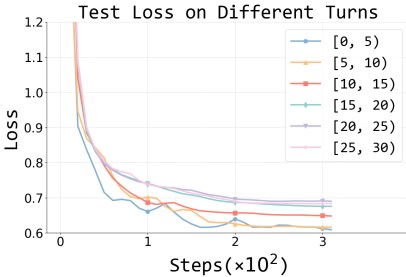

Figure 8: Averaged test loss on different turns of training data.

## 5 RELATED WORK

In the course of deep learning's development, several intriguing training dynamics have gradually been discovered, such as grokking (Power et al., 2022), double descent (Belkin et al., 2019), emergent abilities (Wei et al., 2022), and zero-shot generalization on unseen tasks. The first three phenomena have already received substantial explanations. For grokking, it happens from algorithm tasks (Power et al., 2022) to a broader spectrum of realistic tasks (Liu et al., 2022), explained through the Slingshot Effects (Thilak et al., 2022) or the competition between memorization and generalization circuits (Varma et al., 2023). For double descent, Nakkiran et al. (2021) proposed that double descent occurs both model-wise and epoch-wise, while Davies et al. (2023) unified grokking and double descent through the learning speed and generalization ability of patterns. Regarding emergent abilities, Schaeffer et al. (2024) suggested that the occurrence of emergent abilities is due to researchers using non-smooth metrics. Hu et al. (2023) extended this idea by proposing an infinite resolution evaluation to predict the emergence of abilities, and Du et al. (2024) attempted to understand emergent abilities from the perspective of loss. Research on these dynamics offers valuable insights for investigating zero-shot generalization, especially the work on emergent abilities. We suggest that model-wise emergent abilities may be analogous to step-wise zero-shot generalization.

LLMs have been proven capable of zero-shot generalization across a variety of downstream tasks (Brown et al., 2020), and instruction tuning has emerged as the most effective method to

achieve this (Chung et al., 2024; Sanh et al., 2021; Wei et al., 2021). The zero-shot generalization phenomenon resulting from instruction tuning is crucial for building general LLMs. This means that models trained on certain tasks can generalize well to unseen tasks. Multi-task datasets designed for this purpose have continuously iterated in quality, quantity, and diversity, and numerous studies have explored how zero-shot generalization occurs during the instruction tuning process. Sanh et al. (2021) constructed P3 using explicit multitask learning, demonstrating that explicit task prompt templates can promote zero-shot generalization. Wang et al. (2022) created the Super Natural Instructions V2 (NIV2) dataset, which comprises over 1600 task types, and empirically showed that more observed tasks, an adequate number of training instances, and larger models improve generalization. Meta introduced OPT-IML (Iyer et al., 2022), investigating the impacts of dataset scale and diversity, different task sampling strategies, and the presence of demonstrations on generalization. Subsequently, Longpre et al. (2023) proposed the Flan Collection, which encompasses up to 1836 tasks, and pointed out that scaling the number of tasks and model size, as well as incorporating chain-of-thought data, can dramatically improve performance on unseen tasks.

In addition, a line of research focuses on understanding the relationships in task-pair transfer (Song et al., 2019; Vu et al., 2020; Zamir et al., 2018), suggesting that not all tasks contribute positively to zero-shot generalization; some tasks may even result in negative transfer effects (Kim et al., 2023; Muennighoff et al., 2023; Zhou et al., 2022). However, a significant limitation of the aforementioned works is that, whether training on one task and then evaluating on another (Kim et al., 2023; Zhou et al., 2022), or simply calculating intermediate task (Vu et al., 2020) or instruction (Lee et al., 2024) transfer scores, all these efforts to select the most informative tasks to promote zero-shot generalization are confined within the "task" framework. This approach is based on a premise: the human-defined "tasks" and even "categories" are sufficiently reasonable. This is precisely the issue our study strives to address: breaking free from the task-level framework to explore "generalization" at a more fundamental level.

## 6    CONCLUSION

Our research sheds light on the mechanism underlying zero-shot generalization during instruction tuning, moving beyond the conventional task-level analysis to a more data-centric and fine-grained perspective. By demonstrating that zero-shot generalization occurs early during instruction tuning and is significantly influenced by data similarity and granularity, we provide a new understanding of how instruction tuning brings up zero-shot generalization. Our proposed Test-centric Multi-turn Arrangement further illustrates the importance of accessing high-similarity data early in the training process to facilitate continual learning and loss reduction. For future work, we suggest exploring the quantitative relationship between similarity distance and loss. Specifically, investigating whether similarity distance can predict a model's generalization performance on new data could further help the optimization of instruction tuning. We hope our findings will pave the way for developing more aligned and robust LLMs, enhancing their ability to generalize effectively in diverse applications.

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

| # Instances Per Task | # Instances Per Eval Task | Add Task Name | Add Task Definition | # Pos/Neg Examples | Add Explanation | Tk Instruct |
|---|---|---|---|---|---|---|
| 100 | 100 | False | True | 2/0 | False | False |

Table 1: The hyper-parameters applied in NIV2 configuration.

APPENDIX

## A  DETAILS FOR SECTION 2

### A.1  DATA AND SETTING

We utilized three datasets: Super Natural Instructions V2 Wang et al. (2022), Public Pool of Prompts Sanh et al. (2021) and Flan-mini Ghosal et al. (2023). Here, we provide a detailed overview of each dataset.

**NIV2.**  Super Natural Instructions V2 (NIV2) is a large collection of tasks and their natural language definitions/instructions, with 746 tasks comprising a total of 74,317 instances in train split. In the NIV2 dataset, each task is characterized by its task name, task definition, positive examples, negative examples, and explanations, accompanied by several task instances comprising input and output. We adopt the default configuration in NIV2 repository [1], as illustrated in Table 1.

**P3.**  Public Pool of Prompts (P3) is a collection of prompted English datasets covering a diverse set of NLP tasks. It is organized into a three-level hierarchy of category, task, and prompt template. For each task, instances are organized into a group of data according to several prompt templates. We refer to such binary pairs of (task, prompt) as a base-class dataset. We utilize the same base-class datasets for training and evaluation as we did for training and evaluating vanilla T0 [2]. In the end, we filter out 284 training base-class datasets and 123 evaluation base-class datasets. Due to the vast amount of the P3 dataset and preliminary experiments indicating early zero-shot generalization, for each training base-class dataset, we randomly select up to 100 instances, resulting in a total of 28,372 training instances.

**Flan-mini.**  The flan-mini dataset is a carefully selected subset maintaining a high level of task diversity while reducing the overall FLAN collection size, encompassing not only the Flan2021 Collection and P3 data but also various ChatGPT datasets, including Alpaca, Code Alpaca, and ShareGPT, significantly increasing the diversity of tasks in the flan-mini dataset. In total, there are 1825 tasks, with 1600 tasks allocated for training and 225 unseen tasks for evaluation. Due to the vast amount of training data and preliminary experiments indicating early zero-shot generalization, we randomly select up to 20 instances for each training task, resulting in a total of 28,751 training instances.

### A.2  TRAINING TEMPLATE AND EXAMPLES

Concatenating the various fields from the data, examples of complete training data appear as follows:

*NIV2 example*

```
User: Definition: In this task, you will be shown a sentence, and you
should determine whether it is overruling or non-overruling. In law, an
overruling sentence is a statement that nullifies a previous case
decision as a precedent by a constitutionally valid statute or a decision
 by the same or higher ranking court which establishes a different rule
on the point of law involved. classify your answers into overruling or
non-overruling.
```

---

[1] https://github.com/yizhongw/Tk-Instruct/blob/main/scripts/train_tk_instruct.sh

[2] https://huggingface.co/bigscience/T0pp

| Model | Max Length | Epochs | BS Per Device | LR | Save Steps | LR Scheduler | Optimizer |
|-------|-----------|--------|---------------|-----|-----------|--------------|-----------|
| LLaMA-2-7B | 1024 | 1 | 8 | 1e-06 | 10 | Cosine | AdamOffload |

Table 2: The hyper-parameters applied during the instruction tuning. *LR* denotes the learning rate and *BS* denotes the batch size.

```
 Positive Example 1 -
Input: 876 f.3d at 1306.
Output: non-overruling.

 Positive Example 2 -
Input: we disapprove cooper and craven to the extent that they may be
read to conflict.
Output: overruling.

Now complete the following example -
Input: the court's discussion fails to adequately account for the origin
of the specific intent element that both section 2(a) and 2(b) contain.
Output:
Assistant: non-overruling.
```

*P3 example*

```
User: I took part in a little mini production of this when I was a
bout 8 at school and my mum bought the video for me. I've loved it ever
since!! When I was younger, it was the songs and spectacular dance
sequences that I enjoyed but since I've watched it when I got older, I
appreciate more the fantastic acting and character portrayal. Oliver Reed
 and Ron Moody were brilliant. I can't imagine anyone else playing Bill
Sykes or Fagin. Shani Wallis' Nancy if the best character for me. She put
 up with so much for those boys, I think she's such a strong character
and her final scene when... Well, you know... Always makes me cry! Best
musical in my opinion of all time. It's lasted all this time, it will
live on for many more years to come! 11/10!! How does the reviewer feel
about the movie?
Assistant:  They loved it
```

*Flan-mini example*

```
User: Do these sentences have the same meaning?
" The bank requires growth from elsewhere in the economy and needs the
economy to rebalance , " he said in an interview with the Press
Association news agency .
The Bank of England " requires growth from elsewhere in the economy and
needs the economy to rebalance , " he told the Press Association news
agency .

Available options:
 (1). no;
 (2). yes;
Assistant:  (2).
```

## A.3 HYPER-PARAMETER DETAILS

For instruction tuning, we present some key hyper-parameters related to instruction tuning in Table 2. Additionally, we utilize the model-center framework (modelcenter, 2023) to conduct full-parameter instruction tuning of LLaMA-2-7B on two 80GB A800s and dynamically adjust the loss scale based

| Model | Max Gen Length | Repetition Penalty | Batch Size | Top-p | Temperature |
|---|---|---|---|---|---|
| LLaMA-2-7B | 128 | 1.2 | 8 | 0.9 | 0.9 |

Table 3: The hyper-parameters applied during the generation.

on the changing training loss to prevent underflow. All of our instruction tuning experiments utilize these hyper-parameters consistently.

For generation, we present some key hyper-parameters during the generation in Table 3. We still employ the model-center framework to conduct the generation of LLaMA-2-7B on one 80GB A800. All of our generations utilize the aforementioned hyper-parameters consistently.

## A.4 EVALUATION DETAILS

**Instruction-tuned model as a generalist.** Initially, we evaluate the model's generalization ability at a holistic level, termed as a generalist. To achieve this, we randomly select 120 samples from all testing data, including a series of unseen tasks. These samples are evaluated against a series of fine-grained checkpoints saved during the instruction tuning stage. The average scores for Loss, ROUGE-1, ROUGE-L, RM Score, and Exact-Match across all samples are calculated. We present the calculation details and formulas of each metric above.

- **Loss**: We use cross entropy to calculate the error between labels and predictions. Because the position with a value of -100 in labels is a padding position, we ignore the prediction at this position during calculation.
- **ROUGE-1**: ROUGE-1 measures the comprehensiveness of the generated summary by calculating the overlap between words in the generated summary and words in the reference summary:

$$\text{ROUGE-1} = \frac{\text{Number of overlapping unigrams}}{\text{Total number of unigrams in reference summary}} \tag{1}$$

- **ROUGE-L**: ROUGE-L is based on the idea of Longest Common Subsequence (LCS). By measuring the length $m$ of the reference summary $X$ and the length $n$ of the generated summary $Y$, the ROUGE-L score is calculated as follows:

$$R_{lcs} = \frac{\text{LCS}(X,Y)}{m} \tag{2}$$

$$P_{lcs} = \frac{\text{LCS}(X,Y)}{n} \tag{3}$$

$$F_{lcs} = \frac{\left(1 + \beta^2\right) R_{lcs} P_{lcs}}{R_{lcs} + \beta^2 P_{lcs}} \tag{4}$$

- **Exact-Match**: First of all, we normalize the answers by removing extra spaces, removing punctuation, and converting all characters to lowercase. Then, for each question-answer pair, if the characters of the model's prediction exactly match the characters of the true answer, EM = 1, otherwise EM = 0. This is a strict all-or-nothing metric; being off by a single character results in a score of 0.

$$\text{EM} = \begin{cases} 1 & \text{if the model's prediction exactly matches the true answer} \\ 0 & \text{otherwise} \end{cases} \tag{5}$$

- **RM score**: Let $S$ represent the sentence to be evaluated, $f$ is the reward model function, which takes as input the sentence $S$ and model parameters $\theta$. We use UltraRM-13B as the reward model. Formally, the RM score assigned to sentence $S$ is defined as:

$$R(S) = f(S, \theta) \tag{6}$$

| | NIV2 | | | P3 | | | Flan-mini | | |
|---|---|---|---|---|---|---|---|---|---|
| | **Train** | **General Eval** | **Speical Eval** | **Train** | **General Eval** | **Speical Eval** | **Train** | **General Eval** | **Speical Eval** |
| **# Tasks** | 746 | — | 119 | 284 | — | 123 | 1600 | — | 225 |
| **# Instances** | 74317 | 120 | 595 | 28372 | 120 | — | 28751 | 120 | 1121 |

Table 4: Detailed statistics for train and test splits of NIV2, P3, and flan-mini in our experiments. *General Eval* denotes the evaluation as a generalist. *Special Eval* denotes the evaluation as a specialist.

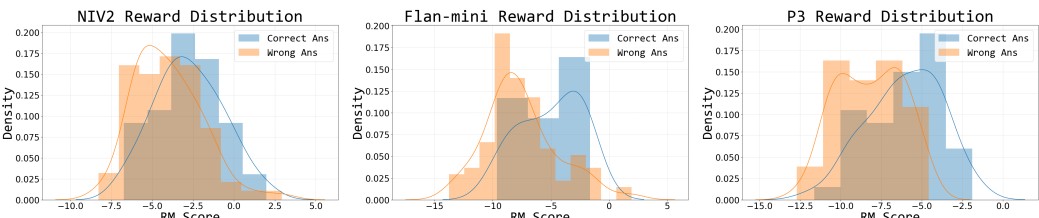

Figure 9: The reward distribution regarding the answer's correctness on NIV2 (left), Flan-mini (middle), and P3 (right). The area under both curves in each figure has large overlaps, indicating the reward cannot well distinguish the quality of answers.

**Instruction-tuned model as a specialist.** In order to facilitate more granular research on task-level scenarios, i.e., exploring the model's generalization ability on specific unseen tasks, termed as a specialist, we take NIV2 and flan-mini datasets as examples. For each unseen task, we randomly select up to five testing instances. As shown in Table 4, for evaluation as a specialist, the flan-mini test set comprises a total of 1,121 instances, covering all 225 unseen tasks. Additionally, the NIV2 test set contains a total of 595 instances, covering all 119 unseen tasks.

Subsequently, we allow a series of fine-grained checkpoints to generate answers on these 1,121 testing instances and compute the loss. We define the generalization metric on a specific unseen task as the average loss of up to five testing instances for that task, to verify whether the model specializes in it.

## A.5 DISCUSSION FOR METRICS

In previous experiments, we have discovered that zero-shot generalization might occur early in the instruction tuning process based on the ROUGE series, Exact-Match, and RM score. However, these metrics may not be suitable for measuring generalization effectively. First, for the ROUGE series, ROUGE-1 refers to the overlap of unigrams, and ROUGE-L is based on the longest common subsequence. Both metrics are limited to surface-level matching, primarily relying on lexical overlap between model outputs and labels, to the extent that capturing semantic similarity or a deeper understanding of the content conveyed in the sentences becomes challenging Ganesan (2018); Grusky (2023). Outputs and labels with different wordings but similar meanings may receive low ROUGE scores.

While the reliability of ROUGE series scores is questionable, metrics like Exact-Match are nonlinear, and previous research Schaeffer et al. (2024) has shown that nonlinear metrics are prone to observing emergent abilities. Although emergence is a model-wise phenomenon, if we adopt such nonlinear metrics step-wise, i.e., along the training timeline, we might also observe step-wise "emergence" so-called generalization phenomena. This might lead to misjudgments regarding the timing of zero-shot generalization. Therefore, we need to address this issue.

Acknowledging that the Reward Model (RM) is trained on preference data, it is inevitable that there will be a certain loss of ability when generalizing to out-of-distribution (OOD) datasets Singhal et al. (2024). Consequently, scoring on different datasets may not be precise. As shown in Figure 2, RM scores for NIV2 are notably higher than those for the other two datasets, indicating a bias. Further, we compare the reward distribution with respect to the correctness of the model response respectively on the three datasets. The large overlap under both curves in Figure 9 indicates that RM could not well distinguish between responses of lower or higher quality. This further highlights the inappropriateness of using RM score to reflect zero-shot generalization.

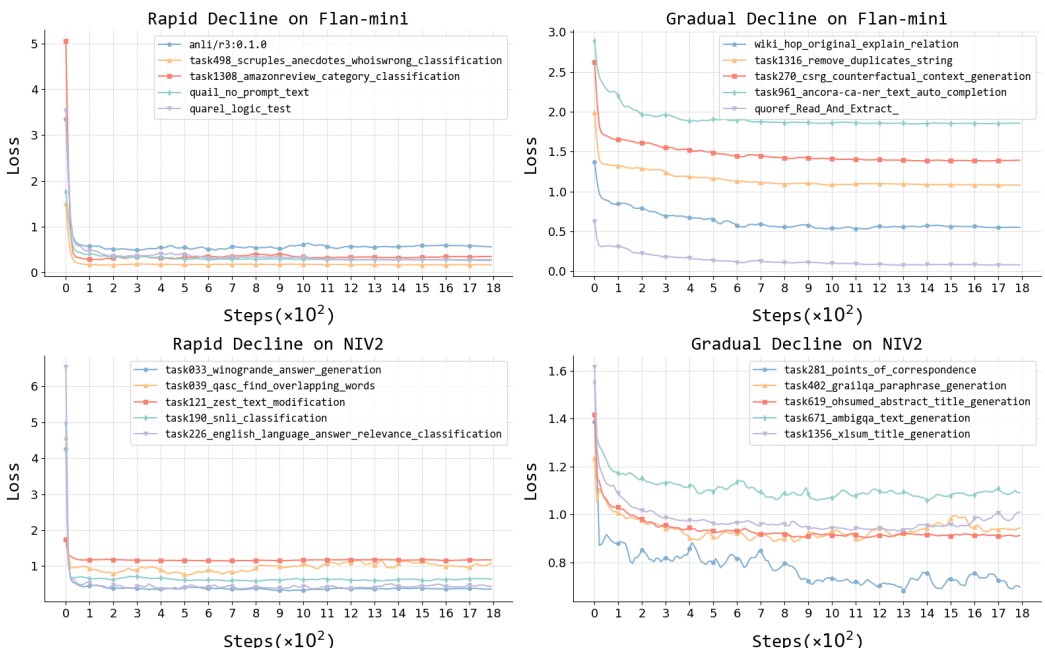

Figure 10: Two main loss trends on the flan-mini and NIV2 test sets. These trends are characterized by a rapid decrease followed by stability and a sharp decline followed by a gradual decrease, respectively. Each type of loss trend is exemplified by selecting five tasks for display.

## A.6 CASE STUDY

Continuing our investigation, we further delve into the fine-grained analysis of the generalization capability on individual unseen tasks.

**Settings**    Taking NIV2 and Flan-mini as examples, we curate a maximum of five test data points for each unseen task, consolidating them into a single test set. Similarly, we generate outputs using a series of fine-grained instruction-tuned checkpoints and compute the cross-entropy loss against the labels and average on a per-task level. For detailed evaluation settings, please refer to Appendix A.4.

**Results.**    From the perspective of individual unseen tasks, zero-shot generalization also occurs in the early stage of instruction tuning. However, different tasks exhibit distinct trends in terms of zero-shot generalization. We identified two primary trends: rapid decrease followed by stability and sharp decline followed by a gradual decrease, as shown in Figure 10. This finding further suggests that the majority of unseen tasks are generalized in the early stage of instruction tuning.

## B DETAILS FOR SECTION 3

### B.1 DIFFERENT TRAINING DISTRIBUTIONS

In Section 3, we take the Flan-mini dataset as an example. For each training task, we select a maximum of 20 data points, and for each testing task, we select a maximum of 5 data points. We employed various training data distributions on Flan-mini. Here, we provide detailed explanations of the data arrangements and training specifics.

- **Round-robin**: In the round-robin setting, with a total batch size of 16, we save checkpoints every 10 steps during instruction tuning. Hence, there is a difference of 160 training data points between adjacent checkpoints. Considering the Flan-mini dataset, where we have divided 1600 training tasks, it takes 1600 data points to traverse all training tasks. Therefore, for every 10 checkpoints (every 100 steps), the model completes one pass over all training tasks.

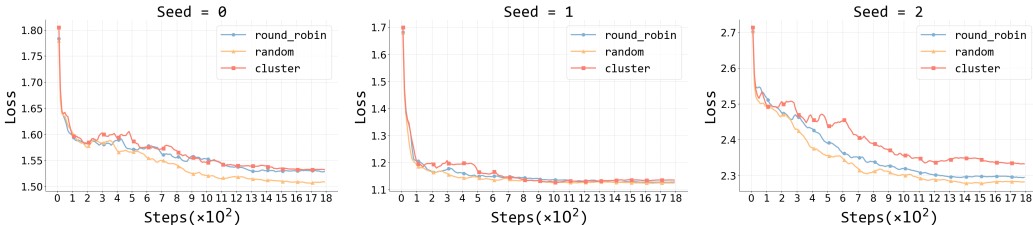

Figure 11: The descent in loss transitions from being gradual ($seed = 2$) to rapid ($seed = 0/1$). (task1264_ted_translation_pl_pt)

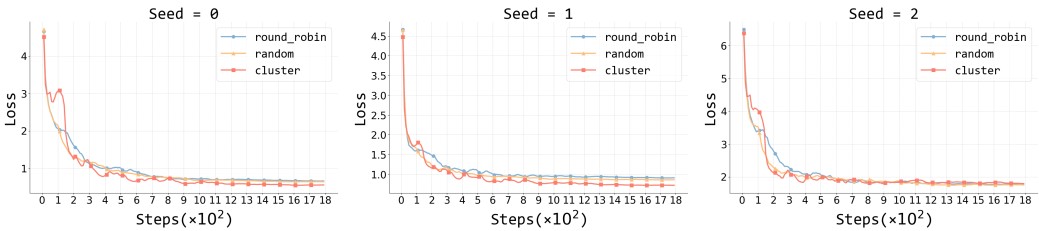

Figure 12: The sudden decrease ($seed = 0/2$) observed in cluster scheduling disappears ($seed = 1$). (task900_freebase_qa_category_classification)

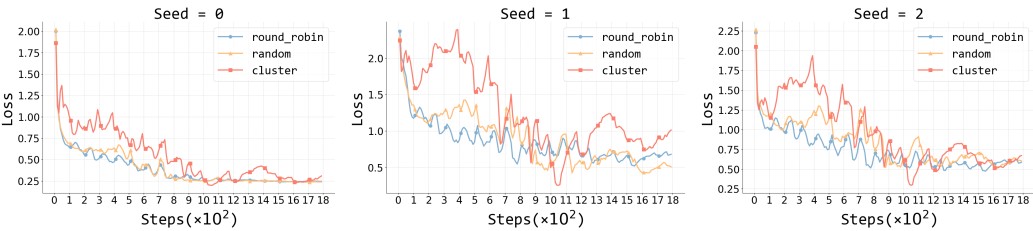

Figure 13: The fluctuation ($seed = 1/2$) in loss becomes more stable ($seed = 0$). (task050_multirc_answerability)

- **Cluster**: In the cluster setting, similarly, there is a difference of 160 training data points between adjacent checkpoints. However, for each training task, we curate a maximum of 20 data points. Consequently, between adjacent checkpoints, the model encounters almost exactly 8 tasks.
- **RT (Random Training)**: As a baseline, we randomly shuffle all training data.
- **NFT (Nearest First Training)**: Given a certain similarity distance measure, we compute the similarity distance from each training data point to the test set based on this measure, and then arrange the training data points from nearest to farthest.
- **FFT (Farthest First Training)**: Given a certain similarity distance measure, we calculate the similarity distance from each training data point to the test set based on this measure, and then arrange the training data points from farthest to nearest.

### B.2 EFFECT OF TEST DATA DISTRIBUTIONS

Upon discovering that controlling the arrangement of training data leads to entirely different loss curves for unseen tasks, we next aim to explore the impact of test data distribution on the results. As the order of test data does not impact the valuation results, we sample test data by employing different seeds to obtain varying test data subsets from the same task. Subsequently, we generate and calculate average loss across a series of fine-grained instruction-tuned checkpoints.

Under different seeds, which represent different subsets of test data for the same task, we observed that the loss curves exhibit distinct patterns:

- The descent in loss transitions from being gradual to rapid (Figure 11).

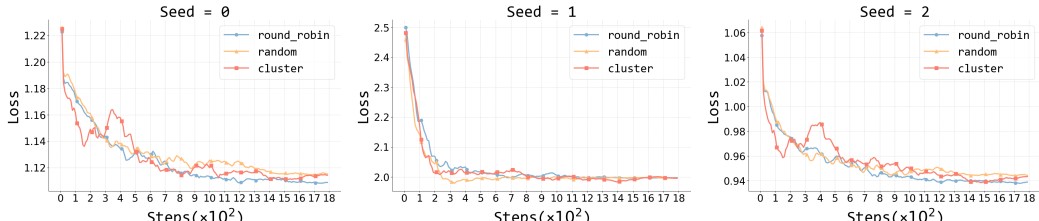

Figure 14: The lowest point of loss shows a significant decrease (from $seed = 1$ to $seed = 0/2$). (task511_reddit_tifu_long_text_summarization)

- The sudden decrease observed in cluster scheduling disappears (Figure 12).
- The fluctuation in loss becomes more stable (Figure 13).
- The lowest point of loss shows a significant decrease (Figure 14).

## B.3 SIMILARITY MEASURE DETAILS

### B.3.1 EMBEDDING-BASED SIMILARITY MEASURE

We utilize all-MiniLM-L6-v2 [1] as our embedding model, which maps sentences to a 384-dimensional dense vector space. When generating the embedding vector of a particular piece of data, we simply format the instruction and answer of this data into a template like "{instruction} {answer}", and then put this whole string into the embedding model to generate the corresponding embedding. After obtaining the embedding of each data, we compute the similarity distance between a training and a test data in the following two ways:

- **Cosine similarity distance**: Cosine similarity determines the similarity by computing the cosine of the angle between the two embeddings, yielding a value between -1 and 1. A value closer to 1 indicates higher similarity, while a value closer to -1 indicates lower similarity. When calculating, we add a negative sign to the cosine similarity to indicate the distance. Thus, a larger value indicates a greater distance between two embeddings. Suppose $\mathbf{A}$ and $\mathbf{B}$ are two embeddings, "·" denotes the dot product of the embedding vectors, and $\|\mathbf{A}\|$ and $\|\mathbf{B}\|$ represent the L2 norms of the embeddings. We calculate the cosine similarity distance as follows:

$$\text{cosine\_similarity\_distance} = \frac{-\mathbf{A} \cdot \mathbf{B}}{\|\mathbf{A}\| \cdot \|\mathbf{B}\|} \tag{7}$$

- **Euclidean similarity distance**: This method calculates the straight-line distance between two points in space. A higher distance value indicates a farther distance between the two embeddings. The Euclidean distance between two points $\mathbf{A}$ and $\mathbf{B}$ in an $n$-dimensional space is computed using the formula:

$$\text{Euclidean\_similarity\_distance} = \sqrt{\sum_{i=1}^{n}(A_i - B_i)^2} \tag{8}$$

Assuming that we have $N_{\text{train}}$ training data and $N_{\text{test}}^T$ test data (for an unseen task $T$), we calculate a similarity distance matrix $D$ with shape $(N_{\text{train}}, N_{\text{test}}^T)$, where each entry $d_{ij}$ represents the cosine or Euclidean similarity distance between the $i^{th}$ training data and the $j^{th}$ test data. For the $k^{th}$ saved checkpoints, it has seen $160 \times k$ training data, so the **S**imilarity **D**istance $SD_k$ between its seen training data and whole test data is calculated using:

$$\text{SD}_k = \text{Op}(D[: 160 \times k][:]) \quad \text{Op} \in [\, min, max, avg, centroid \,] \tag{9}$$

### B.3.2 N-GRAM BASED SIMILARITY MEASURE

During calculation, we still format the instruction and answer of each piece of data in the dataset into a template like "{instruction} {answer}". We then iterate each word in this whole to generate a list of

---

[1]https://huggingface.co/sentence-transformers/all-MiniLM-L6-v2

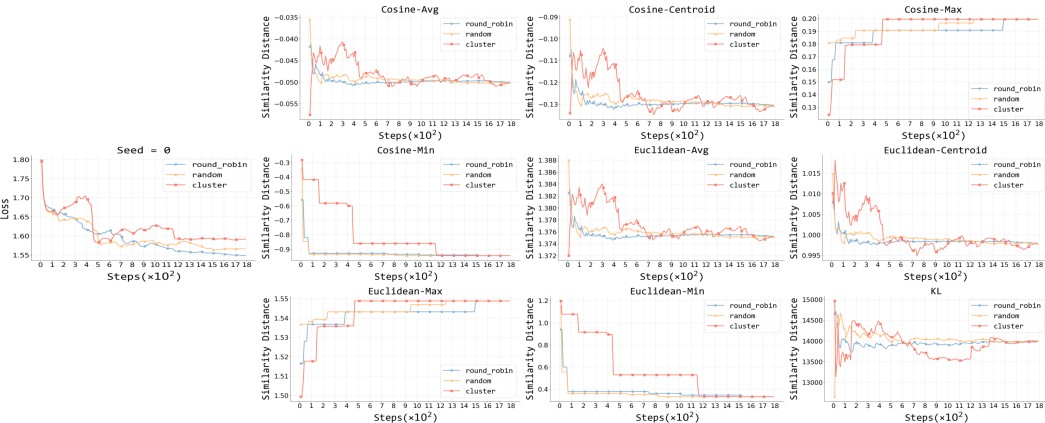

Figure 15: The trends of loss (left) and nine similarity distance measures (right), taking task851 as an example with $seed = 0$.

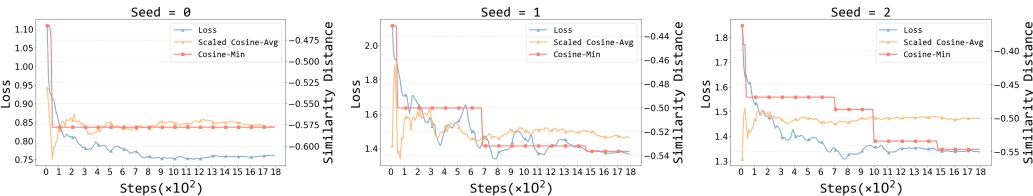

Figure 16: The loss and the similarity distances (scaled cosine average and cosine minimum) between the seen training set and the test set. Since the similarity distances calculated using the minimum (min) metric have a much larger range compared to the average (avg) metric, we consider scaling the average similarity to the same magnitude as the minimum similarity, denoted by "Scaled Cosine-Avg". This will allow for better comparison and analysis between the two metrics.

n-gram tuples, where $n$ represents the length of consecutive words:

$$\text{get\_n-grams}(\text{splice\_data}, n) = \left\{ (w_i, w_{i+1}, \ldots, w_{i+n-1}) \middle| \begin{array}{l} \text{splice\_data} = w_1 w_2 \ldots w_m, \\ 1 \leq i \leq m - n + 1 \end{array} \right\} \quad (10)$$

Then the n-gram tuples of all the data in the dataset are counted, and the frequencies are converted to probabilities to obtain the n-gram distributions of the dataset. Finally, we use KL divergence to represent the similarity distance between two datasets:

- **KL divergence similarity distance**: KL divergence is a measure used to quantify the difference between two probability distributions. When its value is larger, it indicates that the two distributions are less similar. Let $p$ and $q$ represent the probability distributions of training dataset A and test dataset B, where $\epsilon$ denotes the smoothing parameter to avoid division by zero. $p_i$ and $q_i$ represent the probability of the $i^{th}$ n-gram. We compute KL divergence as follows:

$$\text{KL\_divergence\_similarity\_distance}(p, q, \epsilon) = \sum_i p_i \log \left( \frac{p_i}{q_i + \epsilon} \right) \quad (11)$$

So the **S**imilarity **D**istance $\text{SD}_k$ between its seen training data and whole test data is calculated using:

$$\text{SD}_k = \text{KL}(\text{get\_n-grams}(\mathcal{D}_{\text{train}}, N_{\text{train}}), \ \text{get\_n-grams}(\mathcal{D}_{\text{test}}, N_{\text{test}}), \ \epsilon) \quad (12)$$

### B.4 EXPLORING APPROPRIATE SIMILARITY MEASURES

**Setting.** We analyze a series of checkpoints saved during instruction tuning on Flan-mini, based on the three settings described in Section 3.1. We calculate similarity distances between the training data seen by each checkpoint and each unseen task, as depicted in Figure 15. Furthermore, in the

cluster setting, we explore the relationship between a significant decrease in the lowest loss observed with different seeds and the similarity measures. Specifically, suppose there are $N$ instruction-tuned checkpoints and $D$ is the similarity distance matrix, for $k^{th}$ checkpoint in the cluster setting, we compute the Scaled Cosine-Avg and Cosine-Min similarity distances as follows:

$$\text{Cosine-Min}_k = \text{MIN}(D[: 160 \times k][:])$$
$$\text{Cosine-Avg}_k = \text{AVG}(D[: 160 \times k][:])$$
$$\text{Scaled-Cosine-Avg}_k = \text{Cosine-Avg}_k \times \frac{\frac{1}{N}\sum_{k=1}^{N}\text{Cosine-Min}_k}{\frac{1}{N}\sum_{k=1}^{N}\text{Cosine-Avg}_k} \quad (13)$$

**Results.**   We found a strong correlation between the trends of similarity calculated using minimal measure and the trends of loss. In the leftmost plot of Figure 15, we observe sudden drops in both the cluster setting (red) and the similarity distances calculated using the minimal measure around step 450 and step 1150. Interestingly, the magnitude of these drops in similarity distances and loss appears coincidental. Furthermore, in Figure 16, we notice that for $seed = 0$ (left), the Cosine-Min (red) decreases to around -0.58 at approximately 50 steps. In contrast, for $seed = 1$ (middle) and $seed = 2$ (right), the Cosine-Min (red) drops below -0.5 at around 700 steps and 1,000 steps, respectively. Additionally, the lowest loss for $seed = 0$ (left) is significantly lower and exhibits a more stable decrease over time compared to the other two seeds.

Additionally, after carefully examining all 225 unseen tasks, among the nine similarity distance metrics, we observed that the i) fluctuation patterns are almost identical when using Euclidean and cosine distances, as well as when using centroid and average distances; ii) the sudden decrease observed in the loss curve in the preceding subsections seems to coincide with sharp drops when using the minimum distance calculation; iii) the KL divergence does not exhibit a clear pattern of change in relation to the loss, which may be due to the fact that KL divergence calculates differences based on n-gram distributions, without taking into account semantic information; iv) the "max" metric focuses on the least similar data encountered during the instruction tuning process.

For the model during instruction tuning, **Cosine-Avg** reflects the average distance from the seen training set to the test set, providing an overall perspective on the impact of seen samples on the test set. On the other hand, **Cosine-Min** reflects the impact of the closest sample in the seen training set to the test set, providing a local perspective on the influence of seen samples on the test set. Therefore, in the following experiments, we will consider using the **Cosine Average (Cosine-Avg)** and **Cosine Minimum (Cosine-Min)** embedding metrics for similarity calculation.

B.5   PROOF OF OPTIMAL SUBSTRUCTURE PROPERTY

**Property of Similarity Measures.**   Intuitively, we could compute the similarity distance between each training data point and the entire test set, and then reorder the training data based on this similarity distance. In this way, the model encounters the most similar training data point to the test set first during instruction tuning. We demonstrate that this approach exhibits the characteristics of optimal substructure:

**Theorem B.1 (Optimal Substructure of Cosine-Avg and Cosine-Min)** *Let $f$ be a function for calculating dataset-level similarity distance (**Cosine-Avg** and **Cosine-Min**), taking two sets $A$ and $B$ as inputs and outputting a real number. Given a training set $\mathcal{D}_{train}$ and a test set $\mathcal{D}_{test}$, assume $\mathcal{D}_{train}^f$ is obtained by reordering $\mathcal{D}_{train}$ based on the function $f$ in ascending order of similarity distance to $\mathcal{D}_{test}$. For any $i^{th}$ and $j^{th}$ training data $x_i$ and $x_j$ ($i < j$) in $\mathcal{D}_{train}^f$, naturally, we have $f(\{x_i\}, \mathcal{D}_{test}) \leq f(\{x_j\}, \mathcal{D}_{test})$. We also have that,*

$$f(\mathcal{D}_{train}^f[: i], \mathcal{D}_{test}) \leq f(\mathcal{D}_{train}^f[: j], \mathcal{D}_{test}) \quad (14)$$

The characteristic of optimal substructure ensures that the effect of training set arrangement according to Cosine-Avg or Cosine-Min can accumulate over time as more data point is presented to the model.

**Proof of Theorem B.1.**   Let $f$ be a function for calculating dataset-level similarity distance (**Cosine-Avg** and **Cosine-Min**), taking two sets $A$ and $B$ as inputs and outputting a real number. Suppose the

reordered training dataset $\mathcal{D}_{\text{train}}^f$ follows the sequence from the front to the end: $\{x_1, x_2, \cdots, x_i, x_{i+1}, \cdots, x_j, x_{j+1}, \cdots\}$, we consider the unary function $g(i) = f(\{x_i\}, \mathcal{D}_{\text{test}})$, where $i \in [1, 2, 3, \cdots]$. Due to the reordering, the function $g(i)$ is monotonically non-decreasing. We have that:

$$f(\mathcal{D}_{\text{train}}^f[:i], \mathcal{D}_{\text{test}}) \leq f(\mathcal{D}_{\text{train}}^f[:j], \mathcal{D}_{\text{test}}) \tag{15}$$

Firstly, for **Cosine-Avg**, suppose the length of $\mathcal{D}_{\text{test}}$ is $N_{\text{test}}$ we have

$$f(\mathcal{D}_{\text{train}}^f[:i], \mathcal{D}_{\text{test}}) = \frac{1}{i \times N_{\text{test}}} \sum_{p=1}^{i} \sum_{q=1}^{N_{\text{test}}} Cosine(x_p, y_q), \;\; x_p \in \mathcal{D}_{\text{train}}^f, y_q \in \mathcal{D}_{\text{test}} \tag{16}$$

By applying the $g(i)$ function, we have that

$$f(\mathcal{D}_{\text{train}}^f[:i], \mathcal{D}_{\text{test}}) = \frac{1}{i} \sum_{p=1}^{i} g(p) \tag{17}$$

Similarly, we have

$$f(\mathcal{D}_{\text{train}}^f[:j], \mathcal{D}_{\text{test}}) = \frac{1}{j} \sum_{p=1}^{j} g(p) \tag{18}$$

We notice that

$$\begin{aligned}
\frac{1}{j} \sum_{p=1}^{j} g(p) - \frac{1}{i} \sum_{p=1}^{i} g(p) &= \frac{i \sum_{p=1}^{j} g(p) - j \sum_{p=1}^{i} g(p)}{ij} \\
&= \frac{i \sum_{p=i+1}^{j} g(p) - (j-i) \sum_{p=1}^{i} g(p)}{ij} \\
&\geq \frac{i \sum_{p=i+1}^{j} g(i) - (j-i) \sum_{p=1}^{i} g(i)}{ij} \\
&= \frac{i(j-i)g(p) - (j-i)ig(p)}{ij} \\
&= 0
\end{aligned} \tag{19}$$

Similarly for **Cosine-Min**, we have:

$$\begin{aligned}
f(\mathcal{D}_{\text{train}}^f[:j], \mathcal{D}_{\text{test}}) - f(\mathcal{D}_{\text{train}}^f[:j], \mathcal{D}_{\text{test}}) &= \min_{1 \leq p \leq j} g(p) - \min_{1 \leq p \leq i} g(p) \\
&\geq \min_{1 \leq p \leq i} g(p) - \min_{1 \leq p \leq i} g(p) \\
&= 0
\end{aligned} \tag{20}$$

The uses of $\geq$ in the expressions are due to the monotonically non-decreasing property of the $g(i)$ function. Thus, the original expression is proved.

## C  DETAILS FOR SECTION 4

### C.1  DATA

We utilized three datasets: Flan-mini Ghosal et al. (2023), ShareGPT (GPT4) Wang et al. (2023) and NoRobots Rajani et al. (2023). Here, we provide a detailed overview of ShareGPT (GPT4) and NoRobots.

**ShareGPT** ShareGPT contains cleaned and filtered 6k expert conversations generated by GPT-4 used to train OpenChat Wang et al. (2023). We use the version from openchat [1].

**NoRobots** NoRobots is a high-quality English dataset of 10K instructions and demonstrations created by skilled human annotators rather than GPTs. It was modeled after the instruction dataset described in OpenAI's InstructGPT paper Ouyang et al. (2022) and is comprised mostly of single-turn instructions.

Concatenating the various fields from the data, examples of complete training data appear as follows:

---

*ShareGPT example*

```
User: I want you to become my Prompt engineer. Your goal is to help me
 craft the best possible\n prompt for my needs. The prompt will be used
by you, ChatGPT. You will follow the\n following process:\n\n1. Your
first response will be to ask me what the prompt should be about. I will
provide my\n answer, but we will need to improve it through continual
iterations by going through the\n next steps.\n\n2. Based on my input,
you will generate 2 sections. a) Revised prompt (provide your\n rewritten
 prompt. it should be clear, concise, and easily unders]god by you), b)
Questions\n(ask any relevant questions pertaining to what additional
information is needed from me to\n improve the prompt).\n\n3. We will
continue this iterative process with me providing additional information
to you\n and you updating the prompt in the Revised prompt section until
I say we are done.
Assistant: What would you like the prompt to be about?
```

---

*NoRobots example*

```
User: What is the fastest flying bird?
Assistant: The fastest-flying bird is the Peregrine Falcon. When diving,
it has been measured at speeds over 186 miles per hour.
```

## C.2 EXPERIMENTAL SETUP

**Flan-mini.** We randomly selected several tasks as the testing set, while using all the data from the remaining tasks as the training set. Based on the findings in Section 2, which demonstrated that zero-shot generalization occurs early during instruction tuning, we decided to sample around 30,000 data points, maintaining a similar scale to our previous experiments to conserve resources.

**ShareGPT & NoRobots.** We randomly select 200 data points as the testing set, while using all the remaining data points as the training set.

**Settings.** For the three datasets mentioned above, we arrange the training set based on the Test-centric Multi-turn Arrangement. Assuming that we select each turn of training data from the nearest to the farthest, denoted as $\mathcal{D}_{\text{train}}^i (i \in [1, N])$, where $N$ represents the total number of rounds. Similar to the experiments in Section 3, we have also configured the following three settings, while ensuring that the only difference between these three settings is the arrangement of the same dataset:

- **NFT (Nearest First Training)**: We sequentially organize the data for $\mathcal{D}_{\text{train}}^i$ from $i = 1$ to $i = N$.
- **FFT (Farthest First Training)**: We sequentially organize the data for $\mathcal{D}_{\text{train}}^i$ from $i = N$ to $i = 1$.
- **RT (Random Training)**: As a baseline, we randomly shuffle all training data.

## C.3 COMBINING TRAINING DATA FROM DIFFERENT TURNS AT A MORE MACROSCOPIC LEVEL

**Settings.** In the case of Flan-mini, we randomly selected several tasks as the testing set while using all the data from the remaining tasks as the training set. Instead of randomly sampling 30k examples

---

[1]`https://huggingface.co/datasets/openchat/openchat_sharegpt4_dataset`

first, we apply the Test-centric Multi-turn Arrangement method to the whole training data, which exceeded 1 million instances. Assuming that we select each turn of training data from the nearest to the farthest, denoted as $\mathcal{D}_{\text{train}}^i (i \in [1, N])$, where $N$ represents the total number of rounds. We assume that $M$ represents the desired number of training data samples we want to obtain. We considered the following five settings:

- **NFT (Nearest First Training)**: Firstly, we sequentially organize the data for $\mathcal{D}_{\text{train}}^i$ from $i = 1$ to $i = N$ until the accumulated amount exceeds $\frac{M}{2}$, resulting in the dataset $\mathcal{D}_{\text{train1}}$. Subsequently, we proceed to sequentially organize the data for $\mathcal{D}_{\text{train}}^i$ from $i = N$ to $i = 1$ until the accumulated amount exceeds $\frac{M}{2}$, yielding the dataset $\mathcal{D}_{\text{train2}}$. Finally, we merge $\mathcal{D}_{\text{train1}}$ and $\mathcal{D}_{\text{train2}}$ while ensuring that the data is ordered from nearest to farthest.

- **FFT (Farthest First Training)**: We merge $\mathcal{D}_{\text{train1}}$ and $\mathcal{D}_{\text{train2}}$ in the NFT setting while ensuring that the data is ordered from farthest to nearest.

- **RT (Random Training)**: As a baseline, we randomly shuffle all training data.

- **MAX**: We sequentially organize the data for $\mathcal{D}_{\text{train}}^i$ from $i = N$ to $i = 1$ until the accumulated amount exceeded $M$.

- **MIN**: We sequentially organize the data for $\mathcal{D}_{\text{train}}^i$ from $i = 1$ to $i = N$ until the accumulated amount exceeded $M$.

**Results.** Early exposure to highly similar training data is beneficial for generalization while accessing high-similarity data during instruction tuning can facilitate continued learning and further loss reduction. From Figure 17, we find that i) NFT (yellow) and MIN (green) loss curves exhibited nearly identical patterns. This indicates that early exposure to training data that closely resembles the test data is advantageous for generalization. ii) On the other hand, FFT (blue) and MAX (red) loss curves diverged around the halfway point of training (at 950 steps). At this stage, FFT (blue) began encountering training data with high similarity to the test data, resulting in a further decrease in loss. This suggests that accessing high-similarity data during the instruction tuning phase can lead to improved performance.

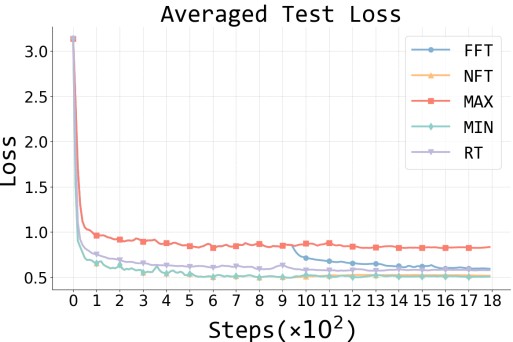

Figure 17: Averaged test loss on five similarity settings under Test-centric Multi-turn Arrangement on Flan-mini.

iii) RT (purple) occupied an intermediate position between the other approaches and served as a baseline. This suggests that RT, which follows a random training strategy, falls between the extremes of NFT and FFT in terms of performance.

### C.4 A Deeper Understanding of Test-centric Multi-turn Arrangement

In the main text, we introduce the **Test-centric Multi-turn Arrangement** method, inspired by transportation theory. In transportation theory, we consider the calculation of the minimum cost required to transform a probability distribution $P(x)$ of a random variable $X$ into another probability distribution $Q(y)$ of a random variable $Y$. This minimum cost is defined as the *Optimal Transport Divergence*, as follows:

$$\text{OT}(P||Q) = \inf_{\gamma \in \Gamma(P,Q)} \mathbb{E}_{(x,y)\sim\gamma}[c(x,y)] \tag{21}$$

where $\Gamma(P, Q)$ denotes the set of all joint distributions $\gamma(x, y)$ whose marginals are $P(x)$ and $Q(y)$, respectively, and $c(x, y)$ represents the cost function measuring the "distance" between $x$ and $y$. A commonly used definition for $c(x, y)$ is the Euclidean distance between two points, which can also be understood as the square of the $L2$ norm. This leads to the definition of the 2-Wasserstein Distance:

$$W_2(P,Q) = \left( \inf_{\gamma \in \Gamma(P,Q)} \mathbb{E}_{(x,y)\sim\gamma}[\|x-y\|^2] \right)^{\frac{1}{2}} \tag{22}$$

More generally, the $k$-Wasserstein Distance is defined as follows:

$$W_k(P,Q) = \left( \inf_{\gamma \in \Gamma(P,Q)} \mathbb{E}_{(x,y)\sim\gamma}[\|x-y\|^k] \right)^{\frac{1}{k}} \tag{23}$$

This definition uses the $k$-th power of the $L2$ norm as the cost function, providing a generalized measure of the "transportation cost" between probability distributions.

In our article, we highlight the significant impact of the similarity between training data and test data on zero-shot generalization. Therefore, a natural question arises: how can we arrange the training data using a better similarity distance measure to achieve better zero-shot generalization? Based on *Optimal Transport Divergence*, we can formalize our problem as follows:

$$\text{Minimize} \sum_{i=1}^{n} \sum_{j=1}^{m} \gamma_{ij} c(x_i, y_j) \tag{24}$$

subject to the constraints:

$$\begin{aligned} \sum_{j=1}^{m} \gamma_{ij} = P(x_i) = \frac{1}{n}, \quad & \forall i = 1, \ldots, n \\ \sum_{i=1}^{n} \gamma_{ij} = Q(y_j) = \frac{1}{m}, \quad & \forall j = 1, \ldots, m \end{aligned} \tag{25}$$

where $\gamma_{ij}$ is the transport plan that minimizes the overall transportation cost between the distributions of the training data $P(x)$ and the test data $Q(y)$. The cost function $c(x,y)$ typically represents the Euclidean distance (L2 norm) between points $x$ and $y$.

The above method treats the distributions $P(x)$ and $Q(y)$ of training and test data as uniform, but this assumption fails when $n$ (training data) and $m$ (test data) are significantly different, causing each training data point to have much less impact compared to each test data point. Hence, we consider treating training and test data equally, with the constraint that the $\Gamma$ matrix contains only 0 or 1 elements. To bridge this gap, we propose the heuristic Test-centric Multi-turn Arrangement method in Algorithm 1 to address the imbalance between training and test data in zero-shot generalization.

This method ensures that each training data point is selected in exactly one round. For the $k$-th round of selected training data $\mathcal{D}_{\text{train}}^k$, for each $x_i$ in $\mathcal{D}_{\text{train}}^k$, there exists a test data point $y_j$ such that $c(x_i, y_j)$ is the $k$-th smallest element in the $j$-th column of the Cost Matrix $\mathcal{C}$ with each entry $c(x_i, y_j)$.

By ensuring this, we achieve a balanced selection of training data points that are optimally distributed according to their similarity to the test data, facilitating more effective zero-shot generalization.

## D  LIMITATIONS

Although our research has made significant progress by discovering that zero-shot generalization occurs in the early stage of instruction tuning and proposing various similarity distance measures to explore their impact on zero-shot generalization, we acknowledge that our study is far from perfect. Firstly, conducting a single experiment can be costly due to storage space requirements and computational resource limitations, so we only conducted limited explorations on a subset of datasets (NIV2, P3, flan-mini) using LLaMA-2-7B with a few runs, which may introduce biases in our conclusions. Secondly, the similarity distance measures we proposed may not have a strong theoretical foundation and can only serve as supplements to existing measures. Lastly, we chose loss as the metric for zero-shot generalization instead of traditional task-level evaluations often seen in

benchmarks like MT-Bench. This is because we believe that traditional task-level generalization has certain limitations, as different tasks or categories may still appear relatively similar to LLMs, while instances from the same task may exhibit profound differences. However, this viewpoint still requires further validation. We hope future works can address these limitations.

# E  COMPUTE RESOURCES

All experiments in this study were conducted on multiple 80GB A800 instances, known for their high-performance capabilities. Conducting full-parameter instruction tuning of LLaMA-2-7B, with approximately 30,000 training data points, utilizing two 80GB A800 instances, required approximately 8 hours. The evaluation pipeline, which involved calculating loss on a series of fine-grained instruction-tuned checkpoints, was executed using a single 80GB A800 instance, taking approximately 10 hours. It is worth noting that the overall research project demanded a greater amount of compute resources than what is solely reported in this paper. This includes additional computational resources utilized for preliminary experiments and unsuccessful attempts that were not included in the final paper.

# F  BROADER IMPACTS

Our work is dedicated to understanding the mechanisms of zero-shot generalization during instruction tuning and proposing several methods to improve zero-shot generalization. This contributes to improving the generalization ability of generative models on unseen tasks. However, it is important to note that these techniques could potentially be utilized for enhancing generalization on harmful tasks as well. Therefore, ethical considerations and responsible deployment of such methods are crucial to ensure their appropriate and beneficial use. Possible mitigation strategies should be conducted, for example, clear policies should be implemented to govern their responsible use while engaging stakeholders to gather diverse perspectives.

