# OpenReview forum: "Improving Zero-Shot Generalization of Instruction Tuning by Data Arrangement"
_ICLR.cc/2025/Conference — ICLR 2025 Conference Withdrawn Submission_

### Official Review · Reviewer_DxiT · 2024-11-01

**Soundness:** 3
**Presentation:** 2
**Contribution:** 2
**Rating:** 5
**Confidence:** 3

**Summary:**

This paper proposes a novel framework, Test-centric Multi-turn Arrangement (TMA), which improves zero-shot generalization in instruction tuning by employing a data arrangement strategy to achieve higher accuracy and enhanced continual learning. The study explores the impact of data similarity and granularity on generalization ability, confirming that introducing highly similar and fine-grained data in the early stages of training can significantly reduce loss. The approach is innovative and attractive.

**Strengths:**

The paper presents a novel framework, Test-centric Multi-turn Arrangement (TMA), which enhances zero-shot generalization in instruction tuning by focusing on the arrangement of training data based on similarity and granularity. Its originality lies in offering a fresh perspective that diverges from traditional task-based methodologies. The research demonstrates high quality through rigorous experimentation and comprehensive evaluations, supported by empirical results.

**Weaknesses:**

The methodology section lacks sufficient clarity regarding the operationalization of the Test-centric Multi-turn Arrangement (TMA), necessitating the inclusion of pseudocode or flowcharts to enhance understanding and reproducibility. Furthermore, the paper does not adequately position TMA within the context of existing literature, particularly in comparison to recent works, which could illuminate its unique contributions.

**Questions:**

1.	The methodology section lacks clarity on the integration of data similarity and granularity within the Test-centric Multi-turn Arrangement (TMA).
2.	Testing primarily on the NIV2, P3, Flan-mini dataset limits the findings' generalizability. Evaluating TMA on additional datasets, would strengthen its applicability.
3.	The reliance on loss as a primary metric should be supplemented with additional metrics (e.g., ROUGE, BLEU) to provide a more comprehensive assessment of performance.

---

### Official Review · Reviewer_8tC4 · 2024-11-01

**Soundness:** 2
**Presentation:** 3
**Contribution:** 2
**Rating:** 5
**Confidence:** 3

**Summary:**

This work examines zero-shot generalization in large language models (LLMs) during instruction tuning, highlighting the limitations of task-based analyses. The authors demonstrate that zero-shot generalization occurs early, with loss as a key indicator. They show that organizing training data by similarity and granularity enhances generalization, especially when models encounter similar and fine-grained data without strict task definitions. The paper introduces a novel method, Testcentric Multi-turn Arrangement, which promotes continual learning and reduces loss. Ultimately, it reveals that zero-shot generalization is a similarity-based process at the instance level, advancing the understanding of LLM alignment.

**Strengths:**

- This paper establishes that zero-shot generalization occurs early in instruction tuning, using loss as a reliable indicator, which can guide future research and practices.

- The investigation into similarity and granularity in data arrangement demonstrates practical strategies for enhancing generalization in LLMs, potentially leading to better performance in real-world applications.

- The proposed Testcentric Multi-turn Arrangement offers an innovative approach to organizing training data, supporting continual learning and effective loss reduction.

**Weaknesses:**

- While the paper provides practical insights, it may lack a comprehensive theoretical framework to fully elucidate the mechanisms underlying the observed effects, which could enhance both understanding and applicability.

- The paper assumes knowledge of test data or data distribution, which may not be feasible for unknown tasks or domains.

- The effectiveness of the proposed data arrangement method may vary significantly based on the specific dataset employed, potentially limiting its generalizability across diverse tasks or domains.

**Questions:**

Pleaser refer to the Weakness.

---

### Official Review · Reviewer_zCKe · 2024-11-03

**Soundness:** 3
**Presentation:** 3
**Contribution:** 2
**Rating:** 5
**Confidence:** 3

**Summary:**

In this paper, the authors explore zero-shot generalization during the instruction tuning process. They observe that such generalization occurs in the early phases, especially when data is arranged in a similar and fine-grained manner. Under these conditions, they find that the model exhibits improved generalization. To support continuous learning and further loss reduction, the authors propose a data arrangement strategy called Test-centric Multi-turn Arrangement.

**Strengths:**

1. This paper is clearly written and easy to follow.

2. Unlike current studies on zero-shot generalization, which primarily focus on integrating diverse tasks or instruction templates, this work is the first to investigate the timing of zero-shot generalization during instruction tuning. This novel approach is quite interesting.

3. The observed improvement in zero-shot generalization during instruction tuning, facilitated by the proposed data arrangement, is notable.

**Weaknesses:**

1. The authors claim that loss is a more consistent measure compared to other metrics such as ROUGE, Exact-Match, and Reward Model scores. However, the analysis in Section 2 does not fully convince me of this claim.

2. Figure 1 could be clearer. In the middle panel, it seems that the yellow data points, which appear similarly in the test set, cause a dramatic decrease in loss, leading to zero-shot generalization. I find the interpretation of this figure confusing, particularly in the context of color and loss changes. Additionally, regarding the bottom figure, does the loss decrease immediately upon encountering the first data point because it shares a colour with the test set? A clearer explanation would be helpful.

**Questions:**

Please see the Weaknesses.

---

### Official Review · Reviewer_zDzi · 2024-11-05

**Soundness:** 3
**Presentation:** 4
**Contribution:** 2
**Rating:** 5
**Confidence:** 3

**Summary:**

This paper investigates the zero-shot generation wrt. instruction tuning. Firstly, the author identifies that zero-shot generalization happens in the early stage of instruction tuning, in line with previous findings. The author proposes to utilize test set loss as a metric to evaluate zero-shot generalization due to its stability and fairness across datasets compared to EM, Rouge and RM. Secondly, the author identifies training data similarity and granularity as two main factors related to efficient zero-shot generalization. Finally, the author proposes TMA for arranging the training data to achieve more effective zero-shot generalization on a target test set.

**Strengths:**

- The paper is well-organized and well-written, very easy to follow.
- The relationship between zero-shot generalization and instruction tuning is a very meaningful research focus with practical benefits.

**Weaknesses:**

- For the data similarity, it is intuitive that train-test data similarity contributes to the zero-shot generalization. For the data granularity, it is directly defined as w/ and w/o task boundary, where breaking the task boundary is for achieving more fine-grained data similarity. It would be better to have more investigation on how to define the "granularity" of data and how it affects the zero-shot generalization.
- Since LLMs are usually general language models, it would be meaningful to consider the zero-shot generalization on more than one task, and whether TMA would harm the generalization on other potential test tasks.
- It would be also good to present the EM/Rouge scores on the test data as a supplement to the conclusion on the loss comparison, since eventually users care about the actual performance e.g., accuracy on the test set.
- The experiments for TMA in Section 4.1 need to compare with other similarity arrangement approaches, such as cosine-avg and cosine-min.

**Questions:**

- In Line 089, which tasks fail to generalize?
- In Figure 3, cluster achieves the lowest loss on task900 and coqa compared to random methods, which seems a little bit contradictory to the conclusion that breaking the task boundary is beneficial to generalization.
- On the study of data granularity to generalization (Section 3.2.2), only two settings are examined. To validate the effect of data granularity, more granularity settings can be compared.

---

### Official Review · Reviewer_B3w7 · 2024-11-05

**Soundness:** 3
**Presentation:** 3
**Contribution:** 2
**Rating:** 3
**Confidence:** 3

**Summary:**

This paper analyzes zero-shot generalization of instruction tuning through the lens of choices of measurement and arrangement of training data. The findings are: (1) test loss can be a good quantitative sign of zero-shot generalization; (2)  zero-shot generalization occurs early during instruction tuning; (3) zero-shot generalization significantly influenced by data similarity and granularity, with higher data similarity early in training and more fine-grained granularity helping zero-shot generalization. With these findings in hand, the authors proposed Test-centric Multi-turn Arrangement that can boost zero-shot generalization further.

**Strengths:**

- The paper is generally well-written.
- The experiments are rigorously designed, leading to sound conclusions.
- The takeaways given in the paper shed light on the process of instruction tuning and how it works.

**Weaknesses:**

- The results shown in the paper are not surprising and a little bit trivial. It is well-known in machine learning that similar samples in training and testing lead to higher test-time performance; also, it has been shown in previous works that a more fine-grained treatment of data instead of some notion of data clusters such as classes or tasks leads to better performance, which is a very natural conclusion. Analysis of unsurprising results is acceptable, but the authors are encouraged to give non-trivial analysis (e.g., a theoretical analysis of why something happens) without just showing the experiment results.

-  All analysis depends on the assumption that the test data is known in advance to make the calculation of data similarity tractable. However, in practice, this assumption does not hold. This makes the proposed algorithm useless.

**Questions:**

- Can you update the introduction (e.g., figure 1) to make the concept of zero-shot generalization and the corresponding analysis easier to understand? I find section 2 and 3 written very good, but the intro should be improved.

---

### Note · Authors · 2024-11-13

I have read and agree with the venue's withdrawal policy on behalf of myself and my co-authors.